# Gaps in cancer care in a multi-ethnic population in Sarawak, Borneo: A central referral centre study

**Melissa Siaw Han Lim**[1,2], **Pei Jye Voon**[3], **Adibah Ali**[4], **Fitri Suraya Mohamad**[2,5], **Lin Lin Jong**[2], **Lee Ping Chew**[2,6,7], **Mohamad Adam Bujang**[7], **Yolanda Augustin**[1,2,8]*, **Yuong Kang Cheng**[2]

1 Department of Paraclinical Sciences, Faculty of Medicine and Health Sciences, Universiti Malaysia Sarawak (UNIMAS), Kota Samarahan, Malaysia, 2 Society for Cancer Advocacy and Awareness Kuching, Kuching, Sarawak, Malaysia, 3 Department of Radiotherapy, Oncology and Palliative Care, Sarawak General Hospital, Ministry of Health, Kuching, Sarawak, Malaysia, 4 Department of General Surgery, Sarawak General Hospital, Ministry of Health, Kuching, Sarawak, Malaysia, 5 Department of Cognitive Sciences, Faculty of Cognitive Sciences and Human Development, Universiti Malaysia Sarawak (UNIMAS), Kota Samarahan, Malaysia, 6 Hematology Unit, Department of Medicine, Sarawak General Hospital, Ministry of Health, Kuching, Sarawak, Malaysia, 7 Clinical Research Centre, Sarawak General Hospital, Ministry of Health, Kuching, Sarawak, Malaysia, 8 St George's University of London, London, United Kingdom

* yaugusti@sgul.ac.uk

**Data Availability Statement:** All relevant data are within the manuscript and its Supporting Information files.

## Abstract

### Background

The state of Sarawak on the island of Borneo in East Malaysia, in working towards developing and strengthening cancer services through a holistic patient-centred approach, must focus on the comprehensive needs of cancer patients by taking into account the psychosocial, cultural and spiritual aspects of Sarawak's multi-ethnic, multi-cultural population.

### Methods

A 42-item survey questionnaire was developed and validated with a total of 443 patients. The perceived importance of information provided and level of patient satisfaction were assessed with a 5-point Likert scale in 10 domains (Diagnosis, Surgery, Radiotherapy, Systemic therapy, Clinical trials, Pain management, Treatment monitoring, Psychosocial support, Sexual care and fertility issues, and Financial support). A Spearman's rank correlation test was applied to determine the correlation between response in both item and domain categories for perceived importance and satisfaction.

### Results

Overall, patients were more satisfied with information related to cancer diagnosis, treatment and surgery but less satisfied with information pertaining to sexual aspects of care and family planning, psycho-social support and financial support. The majority of patients were satisfied with the level of treatment-related information received but preferred the information to be delivered in more easily comprehendible formats. Sexual aspects of care and family planning, psychosocial support and treatment monitoring post-discharge were perceived as

**Funding:** The author(s) received no specific funding for this work.

**Competing interests:** The authors have declared that no competing interests exist.

important but seldom addressed by health care professionals due to lack of professional counsellors, social workers and clinical nurse specialists. Many patients face financial toxicity following a cancer diagnosis, particularly when diagnosed with advanced cancer requiring complex multi-modality treatment.

## Conclusion

Cancer patients in Sarawak have various unmet information needs. Written information and educational videos in local indigenous languages may be more suitable for Sarawak's multi-ethnic population. Sexual aspects of care and family planning are challenging but essential topics to discuss, in particular due to the high prevalence of breast and cervical cancer amongst young women of reproductive age in Sarawak. Financial assessment and information on support services offered by government and non-government organisations should be provided to eligible patients. A holistic needs assessment of each patient at time of diagnosis and support through their cancer journey requires a multi-disciplinary team of medical, nursing and allied health professionals including clinical nurse specialists, pharmacists, counsellors, physiotherapists, occupational therapists, speech and language therapists, dieticians and social workers.

## Introduction

Malaysia is approaching an epidemiologic transition, similar to many developing countries, where cancer diagnosis is on the rise. Sarawak is the largest state in Malaysia spanning 120,000km$^2$ with a population of 2.9 million and is equipped with only one tertiary public hospital (Sarawak General Hospital in the state capital Kuching) with cancer care facilities including surgery, chemotherapy and radiotherapy. Cancer patients from all over Sarawak are referred to Sarawak General Hospital and current facilities are inadequate to cope with the current volume of patients. Sarawak has a multi-ethnic and multi-cultural population consisting broadly of indigenous Iban, Bidayuh, Orang Ulu and Melanau ethnic groups as well as Malay, Chinese, Indian and Eurasian communities. A cancer diagnosis requires a significant level of personal adjustment for patients and their families and can have a profound effect on patient quality of life [1]. It is therefore vital to provide comprehensive patient health literacy and support services to all cancer patients. Accurate, clear and easily comprehensible patient information can alleviate some of the stress associated with a new diagnosis of cancer and empower patients to effectively navigate the cancer management pathway and improve the quality of cancer care [1]. Accurate information may also empower patients to communicate more effectively with family members to support their decision-making process [2]. A previous study reported that good communication between healthcare professionals and patients also requires patient consultation and involvement in terms of understanding patients' ideas, concerns and expectations of cancer care and available support services [3]. Although a number of initiatives have been undertaken by both government and non-government organisations (NGOs) [4] to provide patients in Sarawak with cancer and supportive care information, the effectiveness of these initiatives has not been formally assessed [5].

In supporting patient-centred care [6], each patient should undergo a needs assessment to identify needs that can be addressed by health care professionals as well as allied health professionals and social support groups. In 2011, the International Psychology Society (IPOS)

published its new quality standard, Standard of Quality Cancer Care [7], to support the development and implementation of new clinical practice guidelines. By adopting the quality standard, all cancer patients should receive the most commonly requested information which includes disease-related information but also practical, financial and psycho-social information.

Another study demonstrated that socio-economic status, accessibility to health services, geographical setting, socio-cultural and psychological factors are challenges in addressing different information needs and perceived level of support [5]. In addition, the method of health literacy communication also plays a vital role in influencing the acceptance rate and perceived level of satisfaction. For example, disseminating information using digital media may not be accessible to everyone, particularly for those living in rural areas with reduced digital connectivity. Patients with lower literacy levels may prefer information communicated via video rather than written information. Patients in rural Sarawak often struggle with travel costs and out-of-pocket payments. As a result, some patients turn to non-evidence-based traditional and alternative treatments rather than conventional cancer management.

Whilst various challenges facing Sarawak cancer patients during their cancer journey are frequently reported, there is a need for more in-depth quantitative and qualitative research to better understand the scope and depth of these challenges, in order to provide effective, patient centred solutions. This study aimed to gain insight into the unmet information needs of multi-ethnic cancer patients in Sarawak and to identify barriers and challenges in meeting the cancer care and health literacy needs of these patients. The results of this study are intended to strengthen existing cancer care policy by highlighting current gaps in information provision, communication and supplementary cancer care services in a multi-ethnic population presenting to a tertiary referral cancer centre in the state of Sarawak on the island of Borneo, Malaysia.

## Materials and methods

### Sample and setting

Patients undergoing follow up at the Department of Radiotherapy, Oncology and Palliative Care, Sarawak General Hospital (SGH) were invited to participate in the study. Patients were considered eligible if they were aged 18 years old and above, diagnosed with cancer (all types), treated at SGH and less than five years since their initial cancer diagnosis. Patients with cognitive impairment, those undergoing end-of-life care and those diagnosed with childhood cancers were not eligible for the study. Patients who consented to participate in the study were invited to complete an online questionnaire based on their language preference (English or Malay). Due to the complexity of the questionnaire, a researcher assisted patients with online questionnaire completion. Demographic profiles, socio-economic status and clinical information regarding their cancer diagnosis and treatment were also collected. This study was approved by the Medical Research and Ethics Committee (MREC) of the National Institutes of Health (NIH) Malaysia (REC Reference: NMRR ID-22-00145-TBF (IIR)). Written informed consent was obtained from all participants.

### Questionnaire development

The process of developing the questionnaire involved five phases: (i) to explore and understand the subject matter, (ii) to develop the questionnaire domains and items (iii) to assess content validity and face validity, (iv) to conduct a pilot study and (v) to undertake fieldwork using the questionnaire [8]. The study began with a literature review of published studies related to cancer care. Upon the identification of a list of core components of cancer care, the

authors then developed questionnaire items that were further categorized into ten different domains including diagnosis, surgery, radiotherapy, systemic anti-cancer therapy, clinical trials, pain management, treatment monitoring, psychosocial support, sexual aspects of care and family planning and financial care. The overall aim of the questionnaire was to measure patient perception of importance and satisfaction with each component of cancer care. An interval scale that was based on a five-point Likert scale, i.e., "not important/not satisfied", "slightly important/slightly satisfied", "neutral", "important/satisfied" and "very important/very satisfied", was used in this study.

## Content validity and face validity

The content validity of the questionnaire was assessed by three subject matter experts including two senior physicians and a senior pharmacist. All comments were given due consideration by the researcher, who made further amendments to the questionnaire in order to maintain content validity and the overall accuracy of the content of each item. This was an iterative process, and it was repeated several times until the overall results were considered satisfactory by the subject matter experts. All the questions were initially developed in English. Subsequently, all the questions were translated into the Malay language by using both forward and backward translations. The study group carefully reviewed both the English and Malay versions of the questionnaire, and after making all necessary revisions for both versions of the questionnaire, an assessment of face validity was conducted among a sample group of three patients.

## Pilot study and fieldwork

Once the results based on the assessment of both content validity and face validity were determined to be satisfactory, a pilot study was conducted amongst 10 patients. The aim of the pilot study was to test the reliability of the questionnaire. The Cronbach's alpha of the domains ranged from 0.618 to 0.883. Following the pilot study, the final questionnaire to measure cancer care was developed with ten domains and 42 items.

## Sample size statement

The sample size statement follows a guideline introduced by a previous study [9]. This study aimed to determine the response from patients based on a survey of cancer care. The questions were measured based on a five point Likert-scale and thus the responses analysed in a numerical form, with mean (SD) reported. As the score for each item and domain were presented as a mean value (SD), the sample size calculation was based on a formula to determine a mean value for a targeted population. With an estimated standard deviation of 1.5 and a margin of error of 0.2, the minimum sample size requirement for a 95% confidence interval was calculated as 217 patients. Allowing for a 20.0% refusal/drop-out rate, the required sample size was 272 patients. A correlation test was also conducted between the domains assessing importance and the domains assessing satisfaction. Thus, sample size requirements also needed to be estimated based on a formula from Pearson's correlation test. For the calculation, the correlation coefficient = 0.0 in the null hypothesis versus a minimum correlation coefficient = 0.3 in the alternative hypothesis. Based on a minimum power of 80.0% and type I error of 5.0%, the minimum sample size requirement was 84 patients [10]. To incorporate a 20.0% drop-out, the required sample size was inflated to 105 patients. Since the sample size requirement to estimate the mean for a population was higher, therefore the target sample size requirement for this study was 272 patients.

## Statistical methods

Descriptive analyses were used to present the demographic profiles of patients in the form of frequency, percentage and mean with standard deviation (SD). A Spearman's rank correlation test was applied to determine the correlation between response in both item and domain for importance and satisfaction since the assumption for the parametric test was violated. Patients who had not answered any items were excluded from that domain-specific analysis. All statistical analyses were performed using SPSS version 17.0 (IBM Corp. Released 2017. IBM SPSS Statistics for Windows, Version 25.0, Armonk, NY: IBM Corp.)

## Results

A total of 443 patients were approached from March 2022 to June 2022. Thirty (6.8%) declined to participate, and 20 (4.5%) patients were excluded due to incomplete questionnaires. This led to a final sample of 393 (88.7% response rate). Of these 393 patients, 116 patients (29.5%) responded in English, and 277 (70.5%) responded in Malay.

Table 1 displays the demographic profile of the patients included in the study. The majority were female (69.2%) and mean(SD) age at diagnosis was 54.3 (13.7) years. In terms of ethnicity, Malays accounted for 35.6%, Chinese 30%, Iban 15.3%, Bidayuh 12.7%, Orang Ulu 0.5%, Melanau 3.1% and others 2.8% of the surveyed population. In terms of educational attainment, 7.9% reported not completing formal schooling, 20.1% only completed primary school, 50.1% completed secondary school and 21.9% completed college/university education. Around half of patients surveyed were unemployed (46.8%), 14.8% were employed by the government, 12.2% were employed privately, 7.4% were self-employed, 17.8% were retired and 1% were students. Seventy percent of participants had a monthly household income of less than RM2500, which would classify them as B40 patients (low income earners in the bottom 40% socio-economic category defined by the Malaysian government with a household income of less than RM4500 a month). The majority of patients were married (75.6%), with 12% single, 10.4% widowed and 2% divorced.

Table 2 demonstrates the clinical profile of the patients. More than half (51.4%) of the participants were diagnosed between the age of 50 to 69. The top three most commonly reported cancers at SGH were 1) Breast cancer (36.1%), 2) Nasopharyngeal cancer (11.7%), and 3) Colorectal cancer (11.2%). More than half (57.2%) of the participants were diagnosed with advanced stage 3 and stage 4 cancer (28.2% and 29.0%, respectively). The majority of patients were initially diagnosed less than 6 months prior to questionnaire completion (27.5%) whilst around 23.9% of patients were diagnosed between 3–5 years prior to questionnaire completion.

The most common first type of treatment was surgery (39.2%), followed by systemic anti-cancer therapy (33.3%) and radiotherapy (24.4%). The most common second type of treatment was systemic anti-cancer therapy (31.0%) and radiotherapy (22.6%). Systemic anti-cancer therapy was the most common third type of treatment in patients treated at this centre (20.4%). Overall, pain management therapy was only reported in 11 participants (2.8%). The majority of participants (78.6%) did not have health insurance.

Table 3 demonstrates the importance and satisfaction level of respondents pertaining to information received about various aspects of cancer care. The top 3 domains perceived as "very important" by participants were surgery [mean(SD) = 4.6(0.5)], systemic anti-cancer therapy [mean(SD) = 4.5(0.5)], and radiotherapy [mean(SD) = 4.5(0.5)]. The top 3 domains for which participants reported being "very satisfied" were surgery [mean (SD) = 4.4(0.6)], systemic anti-cancer therapy [mean (SD) = 4.2 (0.6)] and radiotherapy [mean(SD) = 4.2 (0.6)].

**Table 1. Demographic profile of patients.**

| Profile | Category | n (%) |
|---|---|---|
| Gender | Male | 121 (30.8) |
| | Female | 272 (69.2) |
| Age at first diagnosis | 18–29 | 30 (7.6) |
| | 30–39 | 47 (12.0) |
| | 40–49 | 85 (21.6) |
| | 50–59 | 103 (26.2) |
| | 60–69 | 99 (25.2) |
| | 70–79 | 27 (6.9) |
| | 80 and above | 2 (0.5) |
| Ethnicity | Malay | 140 (35.6) |
| | Chinese | 118 (30.0) |
| | Iban | 60 (15.3) |
| | Bidayuh | 50 (12.7) |
| | Orang Ulu | 2 (0.5) |
| | Melanau | 12 (3.1) |
| | Others | 11 (2.8) |
| Education Background | No formal schooling completed | 31 (7.9) |
| | Primary school | 79 (20.1) |
| | Secondary school | 197 (50.1) |
| | College / University | 86 (21.9) |
| Current Employment Status | Employed by Government | 58 (14.8) |
| | Employed Privately | 48 (12.2) |
| | Self-employed | 29 (7.4) |
| | Unemployed | 184 (46.8) |
| | Retired | 70 (17.8) |
| | Full time student | 4 (1.0) |
| Occupational Category | Managerial | 20 (5.1) |
| | Professional | 60 (15.3) |
| | Technician and associate professional | 15 (3.8) |
| | Clerical support worker | 27 (6.9) |
| | Service and sales worker | 25 (6.4) |
| | Skilled agricultural, forestry and fishery workers | 9 (2.3) |
| | Craft and related trades worker | 6 (1.5) |
| | Plant and machine-operator and assembler | 3 (0.8) |
| | Elementary occupation | 7 (1.8) |
| | Army | 4 (1.0) |
| | Others | 217 (55.2) |
| Household Income | < RM2,500 | 278 (70.7) |
| | RM2,501 –RM3,170 | 28 (7.1) |
| | RM3,171 –RM3,970 | 15 (3.8) |
| | RM3,971 –RM4,850 | 10 (2.5) |
| | RM4,851 –RM5,880 | 22 (5.6) |
| | RM5,881 –RM7,100 | 18 (4.6) |
| | RM7,101 –RM8,700 | 5 (1.3) |
| | RM8,701 –RM10,970 | 5 (1.3) |
| | RM10,971 –RM15,040 | 9 (2.3) |
| | >RM15,041 | 3 (0.8) |

*(Continued)*

**Table 1.** (Continued)

| Profile | Category | n (%) |
|---|---|---|
| Marital Status | Married | 297 (75.6) |
| | Single | 47 (12.0) |
| | Divorced | 8 (2.0) |
| | Widowed | 41 (10.4) |
| Health insurance coverage | Nil | 309 (78.6) |
| | Personal private insurance | 41 (10.4) |
| | Company provided private insurance cover | 11 (2.8) |
| | Malaysian Social Security Organization Insurance (SOCSO) | 17 (4.3) |
| | Not applicable (retired persons not eligible for private insurance cover in Malaysia) | 15 (3.8) |

Two domains in which patient perception of importance of that domain appeared more "neutral" were psycho-social support [mean(SD) = 3.8 (0.7)], sexual aspects of care and family planning [mean(SD) = 3.9 (1.1)]. Patients also appeared to be more "neutral" from a patient satisfaction level in terms of information received on sexual aspects of care and family planning [mean(SD) = 3.5 (1.0)], psycho-social support [mean(SD) = 3.7(0.7) and financial care [mean(SD) = 3.8(0.8)].

This study shows that only two items had a higher satisfaction score than importance of statistical significance. Those items are 1) "You were informed about the survival rates for your cancer" [mean(SD) = 3.9 (0.9) vs mean(SD) = 3.7 (1.4) $p < 0.001$]; 2) "You were informed about alternative therapies (e.g. acupuncture, homoeopathy, traditional medicine and therapy)' [mean(SD) = 3.6 (1.0) vs mean(SD) = 2.8 (1.4); $p = 0.018$].

The top two items with the highest difference between perceived importance and satisfaction level of statistical significance were 1) "You were informed about available support groups and counselling services which offer various support to cancer patient, families and caregivers" [mean(SD) = 4.0(1.0) vs mean(SD) = 3.4(1.1); $p = 0.001$] and 2) "You were asked to check the availability of medical insurance coverage by your employer" [mean(SD) = 3.9 (1.1) vs mean (SD) = 3.4(1.0); $p < 0.001$].

Table 4 demonstrates the subgroup analysis for "You were informed about the survival rates for your cancer" of the Diagnosis Domain. On subgroup analysis according to ethnicity, we found that whilst the Malay, Iban, and Bidayuh ethnic groups rated this item 4 or less for both importance [mean(SD); 3.6(1.6), 3.5(1.5) and 3.3(1.5), respectively] and satisfaction [mean(SD); 4.0(1.0), 4.0 (0.9) and 3.9(0.8), respectively], the Chinese ethnic group however scored this item significantly higher in terms of importance versus satisfaction [mean(SD) = 4.2 (0.9) vs mean(SD) = 3.7 (0.9); $p < 0.001$].

Based on the correlation between importance and satisfaction in all domains, all correlations are significant with positive correlation ($p < 0.05$) except for pain management ($p = 0.357$), psychosocial support ($p = 0.260$), sexual aspects of care and family planning ($p = 0.240$), and financial care ($p = 0.057$). The correlations in these domains were not significant due to low correlation coefficients and/or a very small sample size for analysis in certain domains.

## Discussion

The majority of cancer patients in this survey being managed at Sarawak General Hospital were generally satisfied with the level of information received, particularly treatment-related

**Table 2. Clinical profile of patients.**

| Clinical Profile | Category | n (%) |
|---|---|---|
| Type of cancer | Breast | 142 (36.1) |
| | Colorectal | 44 (11.2) |
| | Lung | 33 (8.4) |
| | Stomach | 6 (1.5) |
| | Liver | 2 (0.5) |
| | Nasopharyngeal | 46 (11.7) |
| | Cervix | 22 (5.6) |
| | Ovary | 14 (3.6) |
| | Prostate | 15 (3.8) |
| | Leukemia | 3 (0.8) |
| | Lymphoma | 33 (8.4) |
| | Others | 33 (8.4) |
| Time between questionnaire completion and first diagnosis | Less than 6 months ago | 108 (27.5) |
| | 7 months– 1 year ago | 90 (22.9) |
| | 1 year– 2 years ago | 51 (13.0) |
| | 2 years– 3 years ago | 50 (12.7) |
| | 3 years– 5 years ago | 94 (23.9) |
| Stage of disease at first diagnosis | Stage 1 | 51 (13.0) |
| | Stage 2 | 65 (16.5) |
| | Stage 3 | 111 (28.2) |
| | Stage 4 | 114 (29.0) |
| | Not applicable | 52 (13.2) |
| First type of treatment received | Surgery | 154 (39.2) |
| | Radiotherapy | 96 (24.4) |
| | Systemic anti-cancer therapy | 131 (33.3) |
| | Clinical trial | 1 (0.3) |
| | Pain Management | 2 (0.5) |
| | Others | 9 (2.3) |
| Second type of treatment received | Radiotherapy | 89 (22.6) |
| | Systemic anti-cancer therapy | 122 (31.0) |
| | Clinical trial | 1 (0.3) |
| | Pain Management | 1 (0.3) |
| | Others | 1 (0.3) |
| Third type of treatment received | Systemic anti-cancer therapy | 80 (20.4) |
| | Pain Management | 2 (0.5) |
| Fourth type of treatment received | Pain Management | 5 (1.3) |

information. Most domains were perceived as important to patients, although sexual aspects of care and family planning and psycho-social support domains appeared to be less of a priority for patients (rated below 4). We note that the response rate to the 'Sexual aspects of care and family planning' domain was low (41 responders). The majority of domains received a lower rating for perceived satisfaction compared to perceived importance by patients (Table 3), indicating the need for further development in holistic cancer care services and patient information.

For the diagnosis domain, all items were scored above 4 for perceived importance and satisfaction except for "You were informed about the survival rates of your cancer," where both importance and satisfaction were scored at an average(SD) of 3.7 (1.4) and 3.9 (0.9)

**Table 3. Importance and satisfaction level of respondents pertaining to information received about various aspects of cancer care.**

| Care Aspects | Domain | Mean | n | SD | Mean Differences | corr. coeff | p-value |
|---|---|---|---|---|---|---|---|
| **Diagnosis (n = 362)** | | | | | | | |
| **Diagnosis (overall)** | Importance | 4.298 | 362 | 0.570 | 0.175 | 0.387 | <0.001 |
| | Satisfaction | 4.124 | 362 | 0.576 | | | |
| You were explained clearly why you needed a diagnostic test. | Importance | 4.610 | 376 | 0.584 | 0.309 | 0.557 | <0.001 |
| | Satisfaction | 4.300 | 376 | 0.788 | | | |
| The results of diagnostic test were explained in a way you could understand. | Importance | 4.600 | 374 | 0.548 | 0.265 | 0.587 | <0.001 |
| | Satisfaction | 4.330 | 374 | 0.738 | | | |
| You were given written information about your type of cancer | Importance | 4.160 | 376 | 0.894 | 0.367 | 0.399 | <0.001 |
| | Satisfaction | 3.800 | 376 | 0.978 | | | |
| You were informed about the survival rates for your cancer. | Importance | 3.720 | 372 | 1.431 | -0.194 | 0.232 | <0.001 |
| | Satisfaction | 3.910 | 372 | 0.937 | | | |
| You were informed of the treatment options before any treatment started. | Importance | 4.400 | 377 | 0.789 | 0.149 | 0.576 | <0.001 |
| | Satisfaction | 4.250 | 377 | 0.793 | | | |
| **Surgery (n = 148)** | | | | | | | |
| **Surgery (overall)** | Importance | 4.591 | 148 | 0.455 | 0.188 | 0.568 | <0.001 |
| | Satisfaction | 4.404 | 148 | 0.554 | | | |
| You were informed about the recommended surgery and how it is done. | Importance | 4.650 | 153 | 0.601 | 0.17 | 0.515 | <0.001 |
| | Satisfaction | 4.480 | 153 | 0.64 | | | |
| You were informed about the benefits of the recommended surgery. | Importance | 4.550 | 151 | 0.597 | 0.219 | 0.622 | <0.001 |
| | Satisfaction | 4.330 | 151 | 0.746 | | | |
| You were informed about the possible complications and other expected risks from the surgery. | Importance | 4.560 | 150 | 0.607 | 0.167 | 0.609 | <0.001 |
| | Satisfaction | 4.390 | 150 | 0.759 | | | |
| You were informed about post-operative care and expected time of recovery. | Importance | 4.640 | 151 | 0.583 | 0.205 | 0.591 | <0.001 |
| | Satisfaction | 4.430 | 151 | 0.796 | | | |
| **Radiotherapy (n = 178)** | | | | | | | |
| **Radiotherapy (overall)** | Importance | 4.496 | 178 | 0.502 | 0.285 | 0.508 | <0.001 |
| | Satisfaction | 4.211 | 178 | 0.606 | | | |
| You were informed about the possible benefits of radiotherapy. | Importance | 4.420 | 181 | 0.731 | 0.26 | 0.563 | <0.001 |
| | Satisfaction | 4.160 | 181 | 0.908 | | | |
| You were informed about the potential side effects caused by radiotherapy and how to cope with them. | Importance | 4.540 | 180 | 0.62 | 0.233 | 0.608 | <0.001 |
| | Satisfaction | 4.310 | 180 | 0.777 | | | |
| You were informed about the duration of the entire radiotherapy. | Importance | 4.620 | 181 | 0.581 | 0.099 | 0.690 | <0.001 |
| | Satisfaction | 4.520 | 181 | 0.592 | | | |
| You were informed about who/where to contact regarding any problems you may have after you leave the centre. | Importance | 4.400 | 183 | 0.755 | 0.519 | 0.404 | <0.001 |
| | Satisfaction | 3.880 | 183 | 1.127 | | | |
| **Systemic therapy (n = 320)** | | | | | | | |
| **Systemic anti-cancer therapy (overall)** | Importance | 4.531 | 320 | 0.494 | 0.299 | 0.541 | <0.001 |
| | Satisfaction | 4.231 | 320 | 0.647 | | | |
| You were informed about the possible benefits of systemic therapy. | Importance | 4.480 | 329 | 0.681 | 0.237 | 0.632 | <0.001 |
| | Satisfaction | 4.240 | 329 | 0.826 | | | |
| You were informed about the potential side effects caused by systemic therapy and how to cope with them. | Importance | 4.530 | 324 | 0.679 | 0.259 | 0.618 | <0.001 |
| | Satisfaction | 4.270 | 324 | 0.813 | | | |
| You were informed about the treatment duration of the entire systemic therapy. | Importance | 4.530 | 326 | 0.65 | 0.199 | 0.630 | <0.001 |
| | Satisfaction | 4.330 | 326 | 0.752 | | | |
| You were informed about who/where to contact regarding any problems you may have after you leave the centre. | Importance | 4.590 | 327 | 0.639 | 0.477 | 0.451 | <0.001 |
| | Satisfaction | 4.110 | 327 | 1.039 | | | |

*(Continued)*

**Table 3.** (Continued)

| Care Aspects | Domain | Mean | n | SD | Mean Differences | corr. coeff | p-value |
|---|---|---|---|---|---|---|---|
| **Pain management (n = 9)** | | | | | | | |
| **Pain management (overall)** | Importance | 4.222 | 9 | 0.655 | 0.194 | 0.349 | 0.357 |
| | Satisfaction | 4.028 | 9 | 0.755 | | | |
| You were informed about the need for pain relief medication | Importance | 4.910 | 11 | 0.302 | 0.909 | 0.106 | 0.756 |
| | Satisfaction | 4.000 | 11 | 1.183 | | | |
| You were informed about the potential side effects of pain relief and how to cope with them. | Importance | 4.360 | 11 | 1.206 | 0.364 | 0.354 | 0.299 |
| | Satisfaction | 4.000 | 11 | 1.265 | | | |
| You were informed on the frequency you can repeat the pain medication for extra pain. | Importance | 4.440 | 9 | 1.014 | 0.000 | 0.619 | 0.075 |
| | Satisfaction | 4.440 | 9 | 1.014 | | | |
| You were informed about non-medication approach for pain management. | Importance | 3.000 | 11 | 1.483 | -0.545 | 0.039 | 0.911 |
| | Satisfaction | 3.550 | 11 | 0.82 | | | |
| **Treatment monitoring (n = 377)** | | | | | | | |
| Treatment monitoring (overall) | Importance | 4.462 | 377 | 0.542 | 0.323 | 0.446 | <0.001 |
| | Satisfaction | 4.140 | 377 | 0.707 | | | |
| You were informed about the purpose of the test(s). | Importance | 4.480 | 387 | 0.672 | 0.279 | 0.555 | <0.001 |
| | Satisfaction | 4.200 | 387 | 0.845 | | | |
| You were informed about when you can expect the results. | Importance | 4.320 | 380 | 0.772 | 0.271 | 0.520 | <0.001 |
| | Satisfaction | 4.040 | 380 | 0.863 | | | |
| You were explained about the result of the test and its explanation in a way you understand. | Importance | 4.600 | 381 | 0.583 | 0.415 | 0.502 | <0.001 |
| | Satisfaction | 4.190 | 381 | 0.959 | | | |
| **Psychosocial support (n = 367)** | | | | | | | |
| Psychosocial support (overall) | Importance | 3.806 | 367 | 0.715 | 0.121 | 0.059 | 0.260 |
| | Satisfaction | 3.685 | 367 | 0.715 | | | |
| You were informed about the psycho-social problems caused by the types of cancer you have and its treatment(s). | Importance | 4.070 | 385 | 0.952 | 0.294 | 0.382 | <0.001 |
| | Satisfaction | 3.770 | 385 | 0.929 | | | |
| You were informed about available support groups and counselling services which offer various support to cancer patient, families and caregivers. | Importance | 3.990 | 384 | 0.996 | 0.635 | 0.170 | 0.001 |
| | Satisfaction | 3.350 | 384 | 1.054 | | | |
| You were informed about alternative therapies (eg acupuncture, homeopathy, traditional medicine and therapy). | Importance | 2.810 | 376 | 1.421 | -0.798 | -0.121 | 0.018 |
| | Satisfaction | 3.600 | 376 | 0.974 | | | |
| You were informed about any dietary changes that is required during your treatment(s). | Importance | 4.370 | 380 | 0.869 | 0.353 | 0.508 | <0.001 |
| | Satisfaction | 4.020 | 380 | 0.993 | | | |
| **Sexual aspects of care and family planning (n = 41)** | | | | | | | |
| Sexual aspect (overall) | Importance | 3.921 | 41 | 1.06 | 0.390 | 0.188 | 0.240 |
| | Satisfaction | 3.531 | 41 | 0.989 | | | |
| You were informed about the effect of treatment on your sexuality or fertility. | Importance | 4.210 | 120 | 1.052 | 0.492 | 0.422 | <0.001 |
| | Satisfaction | 3.720 | 120 | 0.98 | | | |
| You were informed about whether you could or could not have sexual relations during treatment. | Importance | 4.080 | 104 | 1.103 | 0.500 | 0.114 | 0.247 |
| | Satisfaction | 3.580 | 104 | 1.049 | | | |
| You were informed about the options that can help in your fertility (eg sperms banking, ovum and embryo storage). | Importance | 3.780 | 55 | 1.343 | 0.473 | 0.225 | 0.099 |
| | Satisfaction | 3.310 | 55 | 1.12 | | | |
| You were informed about when you can start a family or have children after the cancer treatment. | Importance | 3.840 | 55 | 1.316 | 0.291 | 0.363 | 0.006 |
| | Satisfaction | 3.550 | 55 | 1.136 | | | |
| **Financial care (n = 131)** | | | | | | | |
| Financial (overall) | Importance | 4.239 | 131 | 0.765 | 0.489 | 0.166 | 0.057 |
| | Satisfaction | 3.751 | 131 | 0.815 | | | |

(*Continued*)

**Table 3.** (Continued)

| Care Aspects | Domain | Mean | n | SD | Mean Differences | corr. coeff | p-value |
|---|---|---|---|---|---|---|---|
| You were informed about the cost of treatment. | Importance | 4.210 | 384 | 1.06 | 0.206 | 0.310 | <0.001 |
| | Satisfaction | 4.010 | 384 | 0.982 | | | |
| You were informed about access to available resources for financial support. | Importance | 4.070 | 375 | 1.14 | 0.317 | 0.093 | 0.071 |
| | Satisfaction | 3.750 | 375 | 1.133 | | | |
| You were asked to check the availability of medical insurance coverage by your employer. | Importance | 3.900 | 134 | 1.139 | 0.537 | 0.387 | <0.001 |
| | Satisfaction | 3.360 | 134 | 0.992 | | | |

respectively; (p < 0.001), although the score for satisfaction was significantly higher than the score for importance. This result differs with previous studies [11–14] where survival outcome was critical to patient decision-making, especially if considering treatments with variable response rates and toxicity profiles. Our study participants reported that information about survival rates was not as vital to them after a cancer diagnosis. However, on subgroup analysis according to ethnicity, we found that whilst the Malay, Iban, and Bidayuh ethnic groups rated this item 4 or below for both importance and satisfaction, the Chinese ethnic group scored this item significantly higher for importance than satisfaction [mean(SD) = 4.2 (0.9) vs mean(SD) = 3.7 (0.9); p < 0.001] (Table 4). These findings suggest that in a multi-ethnic, multi-cultural setting such as Sarawak, various beliefs and cultural perceptions may shape the perceived value of these domains amongst different ethnic groups. For indigenous ethnic groups such as the Iban and Bidayuh, perceived information about survival was rated as less important, possibly due to cultural perceptions of 'fate' and 'destiny' also reported in other cultures [15, 16]. A similar reason may explain why the Sarawak Malay ethnic group did not place a high importance on cancer survival rates, however this needs further exploration.

In the Diagnosis domain, a statistically significant difference between importance and satisfaction was seen for the item "You were given written information about the type of cancer you had" [mean (SD) = 4.16 (0.8) vs mean (SD) = 3.8 (0.01)] (Table 3). As most patients (78.1%) reported secondary school as their highest educational attainment level, written information about their cancer and treatment in the form of leaflets would likely aid their understanding of their disease. Moreover, the different ethnic groups in Sarawak (Table 1) may benefit from written information being translated into local languages such as Iban, Bidayuh, Orang Ulu and Bahasa Sarawak. This need is substantiated by item "You received an explanation about the result of the test in a way you understand" [mean(SD) = 4.6 (0.6) vs mean(SD) = 4.2(1.0); p < 0.001], where the patients reported reduced satisfaction with how results were explained to them during treatment monitoring compared to perceived importance. For those

**Table 4. Subgroup analysis for "You were informed about the survival rates for your cancer" of the Diagnosis Domain.**

| Ethnic Group | Domain | Mean | n | SD | Mean Differences | corr. coeff | p-value |
|---|---|---|---|---|---|---|---|
| Malay | Importance | 3.570 | 137 | 1.566 | -0.420 | 0.153 | 0.075 |
| | Satisfaction | 3.990 | 137 | 0.989 | | | |
| Iban | Importance | 3.470 | 59 | 1.490 | -0.570 | 0.074 | 0.582 |
| | Satisfaction | 4.040 | 57 | 0.865 | | | |
| Bidayuh | Importance | 3.270 | 48 | 1.455 | -0.620 | 0.109 | 0.475 |
| | Satisfaction | 3.890 | 45 | 0.804 | | | |
| Chinese | Importance | 4.210 | 114 | 0.907 | 0.500 | 0.571 | <0.001 |
| | Satisfaction | 3.710 | 109 | 0.946 | | | |

living in rural areas and not having completed formal education (7.9% of the total patients recruited) educational videos may be more suitable in supporting patient health literacy. However digital connectivity in order to access online patient health literacy materials remains a significant barrier to the implementation of this strategy.

For the radiotherapy domain, the highest discrepancy between the rate of importance and satisfaction was for the item "You were informed about who/where to contact regarding any problems you may have after you leave the centre" [mean(SD) = 4.4 (0.8) vs mean(SD) = 3.9 (1.1); p < 0.001]. This result was similar for the systemic anti-cancer therapy domain [mean (SD) = 4.6(0.6) vs mean(SD) = 4.1 (1.0); p < 0.001], indicating a gap in clear instructions for follow up with patients after their surgery, radiotherapy and systemic anticancer therapy. One of the reasons for loss to follow-up could be that the patients living in rural areas far from cancer facilities face difficulties contacting or returning to hospital when faced with an issue post discharge. Tele-oncology, a branch of telemedicine aimed at improving cancer patients' access to care by diminishing the need for long-distance travel to tertiary-level oncology centres [17] is a potential solution in this setting. However digital connectivity will need to be improved across the state and equitable access to patient communication devices (such as smartphones, tablets or computers) will be needed to enable tele-oncology services to be implemented effectively.

In this study, the highest discrepancy between perceived importance and satisfaction was item "You were informed about the need for pain relief medications" [mean(SD) = 4.9 (0.3) vs mean(SD) = 4.0 (1.2); p = 0.756]. The National Cancer Institute defines pain as one of the most common symptoms in cancer patients which can be caused by cancer, cancer treatment or a combination of factors. An extensive review reported that 27.6% of cancer survivors following curative treatment, 32.4% of the patients on active anti-cancer therapy and 51.9% of the patients with advanced, metastatic or terminal disease reported moderate to severe pain [18]. In this study, 114 (29.0%) of the patients were diagnosed with advanced Stage 4 disease. However, only 11(2.8%) reported receiving pain management therapy. There are several potential barriers to effective pain management; a) attitudes, knowledge and skill of health care professionals; b) attitudes and perceptions of patients and the general public; and c) health care system issues and drug accessibility. A local study reported that several patients in the palliative care setting reported fears about being prescribed morphine despite being counselled prior to opioid initiation [19]. Another study in a Malaysian hospital reported that 40% of the patients had a common misconception that using strong opioids may damage their immune system and cause addiction [20]. As the sample size for pain management was small, the data for this domain should be viewed cautiously. Hence, future studies focusing on cancer pain management in Sarawak are warranted.

Non-pharmacological therapies are an important aspect of pain management [21]. However, study participants did not perceive the item "You were informed about available alternative therapies (e.g. acupuncture, homoeopathy, traditional medicine and therapy)' [mean(SD) = 2.8 (1.4) vs mean(SD) = 3.6 (1.0); p = 0.018] of high importance. This item was rated the lowest in perceived importance in this study. This is contrary to the popular belief that most Asian patients prefer traditional complementary and alternative medicine (CAM), which can potentially cause dangerous side effects and drug interactions with conventional systemic anticancer therapy. These complementary treatments include traditional Chinese medicine, vitamin and mineral supplements and antioxidants [22]. In Malaysia, traditional remedies such as "Sabah snake grass", "noni juice", shark cartilage or the alkaline diet for cancer treatment are widely adopted [23]. It was previously reported that Malaysians spend approximately USD500 million on traditional medicines, compared to USD300 million on conventional medicines annually [21]. Another study reported that despite various efforts to promote awareness of breast

cancer, many women choose to seek alternative rather than conventional medical treatment first, as they fear losing their breast may affect their attractiveness to their husbands [13]. In this study, participants may have rated perceived importance for alternative therapy as low due to communication barriers between patients and their health care providers on the use of CAM. Future qualitative studies to explore patients' perception of CAMs in Sarawak would improve understanding of patients' perspectives on CAMs and enable healthcare providers to support patients with accurate knowledge and patient information (for example on potential drug interactions and adverse effects).

Regarding sexual aspects, all items under this domain were rated higher in perceived importance compared to satisfaction (Table 3), consistent with previous studies [23, 24]. Patients may find discussing sexual aspects of care with their healthcare team difficult due to embarrassment and fear of a topic that many Asian populations consider sensitive or taboo. Cultural values and beliefs may also influence patient willingness to discuss sexuality. Health care professionals may also not feel adequately trained or equipped to deal with sexual care issues [24]. Moreover, at SGH, limited space and facilities also lead to several consultations occurring concurrently in the same clinical area, impacting on patient confidentiality. Hence, sensitive topics such as sexual care may be challenging to address during high-volume consultation clinics. However, this subject is particularly important, especially among Malaysian women as many report a high dependency on their husbands (with some ethnic groups legally allowed to have more than one wife), which may be why patients reported that sexual attractiveness is one of the most critical information needs compared to treatment options [25]. Therefore, healthcare professionals should assess patients to ascertain if they have concerns regarding sexual care, identify barriers to communication about sexual issues and tailor information according to individual patient needs. In the United Kingdom, specialised nurses in breast care were identified decades ago as a valuable source of information and support for women with breast cancer [25–28]. Similar training of oncology clinical nurse specialists should be developed in Sarawak and Malaysia.

For the psycho-social support domain, the highest discrepancy between perceived importance and satisfaction was item "You were informed about available support groups and counseling services which offer various support to cancer patient, families and caregivers" [mean(SD) = 4.0(1.0) vs mean(SD) = 3.4(1.1); p = 0.001], indicating that respondents wanted more information about cancer survivor support services. By interacting with professional counsellors and fellow cancer survivors, patients can be empowered, supported and reassured which can relieve anxiety and improve quality of life. This unmet need for psycho-social support has been reported in previous studies [24, 29]. Psycho-social support is vital for emotional, social and role functioning, and this study has shown that an unmet need continues to exist. Training and support should be given to healthcare professionals to recognise anxiety and depression early on so patients can be offered appropriate interventions such as counselling and support. The reduced capacity of professional counsellors at SGH may also be why healthcare professionals do not readily offer this service to patients. Hence, awareness promotion on the availability of patient support groups should be targeted at both patients and healthcare professionals.

Another item worth debating was "You were asked to check the availability of medical insurance coverage by your employer" [mean(SD) = 3.9 (1.1) vs mean(SD) = 3.4 (1.0); p < 0.001]. Financial toxicity, particularly for lower- and middle-income households, is expected to worsen due to economic disruptions caused by the COVID-19 pandemic and rising health costs. The financial burden is high, particularly among those treated with adjuvant chemotherapy [30], where half (51.4%) of the patients in this study received two or more treatment modalities. Economic hardship can be a significant concern and can arise from high

costs for treatment, lack of insurance coverage and loss of income [30, 31]. Research from the landmark 2012–2014 ASEAN Costs in Oncology reported that one in two cancer survivors (51%) in Malaysia are pushed into economic hardship within the first year of being diagnosed, while one in three Malaysian households (33%) fall below the national poverty line [31]. Although Malaysia is a country with Universal Health Coverage (UHC) and medical care is highly subsidised for patients accessing the public health care system, indirect costs arising from the process of diagnosis and treatment can be substantial, including travel costs and out-of-pocket payments [31]. This problem is even more profound in Sarawak where our survey reported that 70.7% have a household income of less than RM2500 (USD 567) and nearly half of patients in this survey (46.8%) were unemployed. This study also showed that 78.6% of patients do not have insurance coverage, indicating their dependency on the public health care system. Cancer treatment costs can impact not only the patient but also the whole family and future generations. Hence, it is crucial to address their information needs relating to the financial costs of cancer treatment and eligibility assessment for financial support through schemes such as the Peka B40 [32] and My Salam [33] health and social protection schemes. Although private health insurance is a way to alleviate cancer treatment related financial toxicity, the majority of our patients cannot afford private insurance. In view of this, the government can drive collaboration with the insurance industry to create awareness and education to empower more Malaysians to have sufficient protection for health expenditure, particularly in cancer care.

## Limitations

This study is the first to examine the needs and perceptions of Sarawakian patients undergoing cancer treatment at SGH. However, several limitations exist–this study is limited to the local population of Sarawak and may not be generalisable to populations from other states and oncology units in West Malaysia. Secondly, it is also limited by selection bias because we could only recruit patients who could afford to commute to SGH, and we could not capture the needs and perceptions of cancer patients living in more remote rural areas of Sarawak who were unable to attend Sarawak General Hospital for follow up. Third, this study is a quantitative study. Hence, it does not explore in-depth patient experiences of health literacy and patient navigation of the hospital system and care pathway. Patient experiences of external non-hospital related services provided by the charitable and NGO sector were not explored in this study. However, the findings from this study will provide a basis to design future in-depth qualitative studies exploring patient needs from culturally diverse and socio-economically challenged backgrounds in order to address these needs more effectively.

## Conclusion

This study indicates numerous unmet needs among cancer patients in Sarawak. Patients in this study were most satisfied with information related to treatment, surgery and diagnosis but were generally least satisfied with the information received pertaining to sexual aspects of care and family planning, psycho-social support and financial care. Time constraints during the health care provider-patient consultation process may compromise adequate information delivery. In Malaysia, there are around 3.6 oncology specialists for every million persons, with the majority (58%) working in private oncology centres. The recommended number of cancer-related specialists is 10 per million population [34]. In addition, the lack of space and inadequate infrastructure have adversely impacted optimum patient-physician communication, especially in safeguarding the confidentiality and privacy of cancer patients. These constraints highlight the importance of considering a more comprehensive multidisciplinary team (MDT)

approach to ensure information needs are met comprehensively among cancer patients in Sarawak. The MDT may consist of clinical nurse specialists, medical social workers, professional counsellors, psychologists, physiotherapists, speech and language therapists, dietitians and pharmacists. Cancer support groups also have an important supportive role to play. Besides this, the development of patient-centered resources such as patient information leaflets, posters and videos in local languages, financial support, patient navigation services and communication tools specific to the local Sarawak population are warranted. With the findings of this study, we aim to drive informed and well-targeted public policy to strengthen cancer care and service development, particularly among ethnically diverse and socio-economically challenged communities in Sarawak.

## Supporting information

**S1 Dataset. SPSS original dataset.**
(SAV)

**S1 Data.**
(SAV)

## Acknowledgments

The authors would like to thank the Director General of Health Malaysia for permission to publish this study and the reviewers for their valuable comments and suggestions on the manuscript. The authors would also like to thank all the clinicians and pharmacists of the Department of Oncology, Radiotherapy and Palliative Care (Dr Choo Yoke Ling, Dr Kho Swee Kiong, Doreen Kiu Kher Lee, Yap Kian Yee) for their contributions to this study and the founding President of the Society for Cancer Advocacy and Awareness Kuching (SCAN), Sew Boon Lui for the inception of this study.

## Author Contributions

**Conceptualization:** Melissa Siaw Han Lim, Pei Jye Voon, Adibah Ali, Fitri Suraya Mohamad, Lee Ping Chew, Yuong Kang Cheng.

**Data curation:** Lin Lin Jong, Mohamad Adam Bujang.

**Formal analysis:** Melissa Siaw Han Lim, Lin Lin Jong, Mohamad Adam Bujang.

**Project administration:** Mohamad Adam Bujang.

**Supervision:** Melissa Siaw Han Lim, Pei Jye Voon, Adibah Ali, Fitri Suraya Mohamad, Lee Ping Chew, Yolanda Augustin, Yuong Kang Cheng.

**Writing – original draft:** Melissa Siaw Han Lim, Pei Jye Voon, Adibah Ali, Fitri Suraya Mohamad, Lin Lin Jong, Lee Ping Chew, Mohamad Adam Bujang, Yuong Kang Cheng.

**Writing – review & editing:** Melissa Siaw Han Lim, Pei Jye Voon, Adibah Ali, Fitri Suraya Mohamad, Lee Ping Chew, Mohamad Adam Bujang, Yolanda Augustin, Yuong Kang Cheng.

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
