## [Decision Letter · Decision Letter 0]

21 Jul 2023

PONE-D-23-12147Gaps in Cancer Care of the multi-ethnic population of Sarawak, BorneoPLOS ONE

Dear Dr. Augustin,

Thank you for submitting your manuscript to PLOS ONE. After careful consideration, we feel that it has merit but does not fully meet PLOS ONE’s publication criteria as it currently stands. Therefore, we invite you to submit a revised version of the manuscript that addresses the points raised during the review process.

We look forward to receiving your revised manuscript.

Kind regards,

Ranjit Kumar Dehury

Academic Editor

PLOS ONE

Journal Requirements:

a) The name of the colleague or the details of the professional service that edited your manuscript.

b) A copy of your manuscript showing your changes by either highlighting them or using track changes (uploaded as a *supporting information* file).

c) A clean copy of the edited manuscript (uploaded as the new *manuscript* file).

5. Please include a separate caption for figure 1 in your manuscript.

**Additional Editor Comments:**

Dear Authors,

The article has merit and require care full reporting. However, it needs to be improved requiring major review with the comments of both the reviewers. The author have to address the comments to ensure scientific communication while handling the review process.

With regards,

Ranjit

Reviewers' comments:

Reviewer's Responses to Questions

**Comments to the Author**

1. Is the manuscript technically sound, and do the data support the conclusions?

Reviewer #1: No

Reviewer #2: Yes

2. Has the statistical analysis been performed appropriately and rigorously? 

Reviewer #1: No

Reviewer #2: Yes

3. Have the authors made all data underlying the findings in their manuscript fully available?

Reviewer #1: Yes

Reviewer #2: Yes

4. Is the manuscript presented in an intelligible fashion and written in standard English?

Reviewer #1: Yes

Reviewer #2: Yes

5. Review Comments to the Author

Reviewer #1: This manuscript reports important summary information from a survey focusing on information provided to cancer patients undergoing follow up at the Department of Radiotherapy, Oncology and Palliative Care of the Sarawak General Hospital (SGH) in the state of Sarawak on the island of Borneo in East Malaysia. It has a number of strengths, including a large and diverse sample of participants from an under-studied region, a comprehensive survey about information provision (and participant satisfaction with this), and an assessment of potential discrepancies between self-reported importance of and satisfaction with a number of areas of information provision as part of their cancer care. In spite of these strengths, there are a number of weaknesses that reduce the potential impact of this work.

1) The survey was designed to assess participant satisfaction with, and perceived importance of, a number of aspects relating to the cancer care they were receiving. However, the summaries that are provided for these questions do not appear to address the topic suggested by the title "Gaps in Cancer Care".

2) When summaries are provided for different domains, drastic differences in sample sizes are apparent. It is not completely clear from the methods what leads to these differences (though a partial description is provided in the statistical analysis section). If non-response is purely driven by participants' choices, these differences are informative to what might be concluded from the findings. More attention should be given to this, both from a view focusing on the description of the methods, as well as from a view about the potential for informative non-response. This could provide key and novel insights (e.g. the use of Western-driven domains in a different, non-Western context).

3) It would have been informative had the demographic (and clinical) characteristics of the study participants been compared to reference values (e.g. all cancer patients seen at the medical center, all patients receiving care at the medical center, the population of the state).

4) There are a number of methodological aspects of the manuscript that would benefit from additional information. These include:

a - More information might have been provided relating to the development and validation of the survey. Reporting a range of Chronbach's alpha values for the different (domain-specific?) internal consistencies is not adequate to enable careful assessment.

b - The reported sample size determination does not appear to align with the analyses that are actually reported in this manuscript.

c - It is not clear to this reviewer that a paired t-test comparing "importance" and "satisfaction" scores for specific questions (and domain averages?) is ideal for assessing the degree of agreement between (or differences in) these two areas of interest. Such an approach requires multiple assumptions (including those about comparability of measurement scales), and it is not clear that these were adequately considered or assessed.

d - Subset analyses are reported in the results, but descriptions of the approaches used were not provided.

5) There are a number of reporting practices that could be improved. Some examples include:

a - Summary values for comparisons of interest are often reported without a corresponding measure of variability (see e.g. 177, 178-179, 181, and many more reported comparisons).

b - Figure 1 draws lines connecting the various domains together, yet there is no clear rationale as to why it might make sense to place them on a continuum. Removal of the smoothed lines is likely the most appropriate option here.

c - Figure 1 does not provide measures of variability for the reported "importance" and "satisfaction" means (or, perhaps more appropriately, for the differences between the two means).

Reviewer #2: 1. Development of questionaries mentioned about self administrative. So the authors needs to explain the educational characteristics of respondents in methodology section. As you mentioned the study population is from divers culture. please explain about how the self reporting survey is administrated?

2. The research topic is reflected disparity of cancer care of multi ethnic population in Sarawak. So the data needs to presents details disparity/difference of cancer care gaps within various ethnic population of Swaawak.

3. Discussion should be structured properly and should be more specific to the study. The authors should also discuss what the results of their study imply and cite and discuss more of relevant sources/articles.

4. Please highlight the key inference of the present study in conclusion section.

5. At the end of the discussion, the limitations and suggestions for future studies should be mentioned separately.

6. PLOS authors have the option to publish the peer review history of their article (what does this mean?). If published, this will include your full peer review and any attached files.

Reviewer #1: **Yes: **V. Shane Pankratz, Ph.D.

Reviewer #2: No

---

## [Author Response · Author response to Decision Letter 0]

4 Dec 2023

We have written a detailed letter of response to the reviewers and uploaded this with our revised manuscript. Thank you.

---

## [Decision Letter · Decision Letter 1]

26 Dec 2023

Gaps in Cancer Care in a multi-ethnic population in Sarawak, Borneo: A central referral centre study

PONE-D-23-12147R1

Dear Dr. Yolanda Augustin,

We’re pleased to inform you that your manuscript has been judged scientifically suitable for publication and will be formally accepted for publication once it meets all outstanding technical requirements.

Kind regards,

Ranjit Kumar Dehury

Academic Editor

PLOS ONE

Additional Editor Comments (optional):

Dear authors,

The manuscript has been improved according to the requirements of the journal. Hence, the article is being accepted. However, the authors have to address the minor issues and technical issues in subsequent stages.

With regards,

Ranjit

Reviewers' comments:

Reviewer's Responses to Questions

**Comments to the Author**

1. If the authors have adequately addressed your comments raised in a previous round of review and you feel that this manuscript is now acceptable for publication, you may indicate that here to bypass the “Comments to the Author” section, enter your conflict of interest statement in the “Confidential to Editor” section, and submit your "Accept" recommendation.

Reviewer #2: All comments have been addressed

Reviewer #3: (No Response)

2. Is the manuscript technically sound, and do the data support the conclusions?

Reviewer #2: Yes

Reviewer #3: Yes

3. Has the statistical analysis been performed appropriately and rigorously? 

Reviewer #2: Yes

Reviewer #3: Yes

4. Have the authors made all data underlying the findings in their manuscript fully available?

Reviewer #2: Yes

Reviewer #3: Yes

5. Is the manuscript presented in an intelligible fashion and written in standard English?

Reviewer #2: Yes

Reviewer #3: Yes

6. Review Comments to the Author

Reviewer #2: Author have addressed all comment and suggestions. I am glad to review the paper and My best wishes to the authors.

Reviewer #3: Mention the objective in the abstract.

What is financial toxicity?

Line 251: Authors discussing data on subgroup analysis but they did not provide any table in the paper.

There is an anomaly in the line no. 253 [mean(SD) 3.61.6), 3.5(1.5) and 3.3(1.5), respectively].

Line no. 306, give space between 'aspectof'.

Line no. 392, remove extra 'this'.

7. PLOS authors have the option to publish the peer review history of their article (what does this mean?). If published, this will include your full peer review and any attached files.

Reviewer #2: No

Reviewer #3: **Yes: **Imteyaz Ahmad

---

## [Editor Report · Acceptance letter]

21 Jun 2024

PONE-D-23-12147R1 

PLOS ONE

Dear Dr. Augustin, 

I'm pleased to inform you that your manuscript has been deemed suitable for publication in PLOS ONE. Congratulations! Your manuscript is now being handed over to our production team.

Kind regards, 

on behalf of

Dr. Ranjit Kumar Dehury 

Academic Editor

PLOS ONE